# Mechatronic Design of a Prototype Orthosis to Support Elbow Joint Rehabilitation

**DOI:** 10.3390/bioengineering9070287

**Published:** 2022-06-29

**Authors:** Jhoan Danilo Arcos Rosero, Daniel Camilo Bolaños Rosero, Luis Fernando Alape Realpe, Andrés Felipe Solis Pino, Elizabeth Roldán González

**Affiliations:** 1Facultad de Ingeniería, Corporación Universitaria Comfacauca—Unicomfacauca, Popayán 190001, Cauca, Colombia; jhoanarcos@unicomfacauca.edu.co (J.D.A.R.); danielbolanos@unicomfacauca.edu.co (D.C.B.R.); lrealpe@unicomfacauca.edu.co (L.F.A.R.); 2Facultad de Ingeniería Electronica y Telecomunicaciones, Universidad del Cauca, Popayán 190003, Cauca, Colombia; 3Fundación Universitaria María Cano, Popayán 190002, Cauca, Colombia; elizabeth.roldan@fumc.edu.co

**Keywords:** elbow injury, rehabilitation prototype, orthosis, mechatronic design

## Abstract

Injuries in the elbow area, such as lateral and medial epicondylitis, are the leading causes of consultation with health specialists. Therefore, this research proposes the mechatronic design of an orthosis with a graphic interface that supports professionals in the rehabilitation of the elbow joint through the execution of flexion–extension and pronation–supination movements. For the development of the rehabilitation prototype, mechatronic design, co-design, and IDEF0 methodologies are used, performing activities such as actuator characterization, simulations, and modeling, among others. Through the execution of a case study in a real environment, the device was validated, where the results suggest a functional and workable prototype that supports the treatment of pathologies in the elbow area through the execution of the mentioned movements, supposing that this is a low-cost alternative with elements to improve, such as the industrial design and new functionalities. The developed proposal shows potential as an economical product that health professionals can use. However, some limitations related to the design and functionalities in the application domain were identified.

## 1. Introduction

The elbow is the region between the upper and lower arm surrounding the elbow joint [1]. This area plays a vital role in the anatomy of the human body, as its main task is to serve as a link between the shoulder and the hand and facilitate the positioning, stabilization, and fine control of the hand [2]. Typically, this area is formed by bony structures, fluid, and soft tissues that confer stability to the joints and facilitate the kinematics between the humerus with the radius and ulna (forearm) [3]. Some activities that widely use the elbow are feeding, hygiene, and work, among others, which denote the importance of this area in the daily life of the human being.

The constant use of the elbow in the daily life of people is one of the leading causes of injuries in this area of the body; however, it is not the only one: other causes of pain can be sprains, strains, and fractures, among others [4]. Multiple structures make up the elbow joint, and, when any of them are damaged or diseased, injuries usually arise that prevent the normal mobility of the elbow or limit the fulfilment of its functions [5]. Usually, the most common elbow injuries are lateral and medial epicondylitis, which are treated by employing therapeutic exercises of flexion–extension and prone–supination that allow for the physical strengthening of the affected area and are non-invasive treatment alternatives that, in most cases, are effective [6,7].

Currently, the use of portable technologies and devices (orthoses or exoskeletons), together with the simultaneous development of computer programs and data analysis, has contributed to creating and facilitating the management of elbow and upper extremity conditions [8,9]. Likewise, the different solutions offered by engineering have made it possible to create a variety of low-cost products that help to improve the population’s quality of life by contributing to the treatment of different conditions. Methodologies such as mechatronic design [10] and co-design [11] have been used to obtain functional products with better synergy than those developed conventionally [12].

A technical proposal is a study by Cruz-Martínez et al. in [13], where they propose an exoskeleton with seven degrees of freedom in which the anatomical movements of the upper torso are emulated, considering a passive rehabilitation approach that is based on the study of the analysis of pathologies and the exercises applied to treat them, achieving an average operational accuracy of 95%. However, the device is not tested in a natural environment with healthcare professionals. Another related research is that of Liu et al. in [14], who propose a portable exoskeleton with five passive degrees of freedom, supporting the rehabilitation of patients with the upper torso and elbow stiffness. The importance of this system lies in the fact that it is a prototype focused on people with cardiovascular accidents who cannot travel to health centers, thus allowing for rehabilitation at home, although health professionals have not evaluated the device. Dindorf and Wos propose a variant of orthosis for elbow rehabilitation in [15] that uses bioelectrical signals from the brain to control the rehabilitative device; for this purpose, it performed a distributed control based on electrical signals, while bi-muscular pneumatic servo motors were used to drive the system. In this study, visual feedback focused on the movement of the orthosis is used, considering the motor functions of the elbow. In [16], they apply a different vision of a rehabilitative exoskeleton for the upper torso, which seeks to help solve pathological elbow tremor, one of the most common movement disorders. For this, one degree-of-freedom orthosis is proposed to suppress this condition based on the control of speed and voluntary movements. The results suggest a tremor power reduction of 99% in an experimental setting, while it has not been validated in a natural environment. Other relevant perspectives in the area are [17,18,19], which show different efforts to help treat different conditions that can affect the elbow through technology, which speaks to the relevance of the topic in the research community. The studies described above are focused on helping to treat different elbow alterations many times without validation by professionals specialized in the area; in addition, as far as the authors know, the mechatronic design of a technological proposal (orthosis) that supports the recovery of mobility using flexion–extension and pronation–supination movements has not been described, so it is a gap in the literature that this research aims to solve.

Therefore, it is considered that the main contribution of this research is to expand the area of knowledge through empirical evidence of a low-cost elbow rehabilitation prototype, built under the guidelines of mechatronic design and co-design methodology, resulting in an orthosis that can assist the stages of recovery of the mobility of the transverse elbow joint to any elbow pathology, through flexion and supination movements.

Finally, this paper is divided into five sections, including this introduction. Next, the materials and methods used in the research are presented, followed by a review of the results found for the proposed prototype. Finally, we conclude the proposal and present future work in the domain.

## 2. Materials and Methods

Among the essential steps to treating different elbows pathologies are mobility exercises that allow for gaining mobility, recovering strength, and gaining proprioception [20]. This speaks of the daily routine and importance of supporting rehabilitation in an area relevant to human anatomy. To support the solution to this problem, the mechatronic development of an orthosis that allows for flexion–extension and pronation–supination exercises was proposed, allowing us to support the recovery of these pathologies. To obtain the functional prototype that solves the exposed problem, the mechatronic design methodology [10] and the hardware and software co-design methodology [21] were followed, which allows for obtaining a synergic product that harmonizes the different areas of knowledge required (electronic design, 3D design, programming, linearization, and sensor characterization). Finally, to validate the results in a real environment, a case study was performed following some guidelines reported in [14]. In the execution of this validation tool, a physiotherapist expert in upper torso pathologies analyzed the contribution of the orthosis to support the recovery of these conditions by using satisfaction surveys, as reported in [22]. Now, among the main materials used in the construction of the proposal were LabVIEW for the graphical interface, Matlab and PAST (PAleontological STatistics) [23] for data processing, and SolidWorks for computer-aided design (CAD), and Arduino as the prototype controller.

A generic scheme with the most important phases of the characterization, design, and implementation of the proposed prototype can be seen in Figure 1.

### 2.1. System Characterization and Design

In this section, the main requirements of the system are specified and characterized. This is because it is a prototype oriented to people in a real environment, so some bibliographic references based on non-technological devices that support the treatment of different elbow conditions are used. For this, bibliographic studies of the Association for the Study of Osteosynthesis (AO) and the American Academy of Orthopaedic Surgeons (AAOS) were used to establish reference precedents to work on the proposal.

The orthoses seek to facilitate the patient’s effective recovery in their therapies, improving the mobility of the elbow joint through the specific graduation of angles and forces as time and recovery phases elapse. These devices currently help patients and physiotherapists to recover joint movements, considering specified ranges [24]. With flexion–extension, the ranges are 0–150${}^{\circ }$, pronation 75${}^{\circ }$, and supination 85${}^{\circ }$. These values help to standardize the limits of elbow rehabilitation [25]. In this sense, the previous data allow for defining reference values for the proposal, besides allowing for the arc modification when the physiotherapist requires it. Finally, Table 1 exposes the mobility values of the elbow.

Some crucial data were analyzed and calculations were designed, such as the arm, forearm, and hand measurements and mass. The values presented in Table 2 correspond to the values obtained based on the proposal of [26].

The prototype comprises three important subsystems for the proposed device: an adjustable subsystem for arm support, a coupling subsystem for the motor, and a grip subsystem for patient pronation and supination. All of the parts were designed in the SolidWorks program considering all the dimensions and data in Table 2. For selecting the material for the mechanisms, physical and mechanical properties were considered, as well as cost, accessibility, and feasibility for machining on CNC (computer numerical control) machines. Figure 2 shows the sequential activities used to develop the final prototype design.

### 2.2. System Modeling

The method known as IDEF0 was used to model the system, which allows for representing a system’s decisions, actions, and activities at both general and specific levels. In addition, it allows for analyzing, communicating, identifying points of improvement, and optimizing processes efficiently and consistently [27]. Figure 3 shows the general system diagram (IDEFA-0), in which, the input, function, mechanisms, and control information governing the system can be identified.

In the prototype’s development and following the parameters of the IDEF0 method, it defined the elbow injury as the primary input (Figure 3). It converted this information to produce the outputs, which are the exoskeleton prototype and the control interface. At the top of the diagram is the control information, which is the information that governs or regulates how and when an activity is performed; for the proposed prototype, these data are the level of technological maturity, movements, arm readings, and continuous monitoring by healthcare specialists. The lower part shows the mechanisms, which are the resources needed to execute the process. In elbow rehabilitation, the general mechanisms are literature review, material study, CAD-CAM design, programming, and validation.

The detailed development of the prototype can be seen in Figure 4, which shows all the processes involved in the execution of the rehabilitation orthosis. Each function or activity exemplifies all of the parameters of the IDEF0 modeling, which have input, control information, mechanisms, and outputs.

Using a methodology such as IDEFA0 made it possible to model and organize the construction of the proposal, acting as a guiding element in the orthosis’s execution, besides providing a functional perspective of the system.

### 2.3. Design and CAD Simulation of the System

Following the mechatronic design methodology [21] with the IDEF0 guide, the computer-aided design was used to visualize the final prototype and simulate the pressures on the proposed prototype. In Figure 5, it observed the CAD design of the support orthosis for elbow rehabilitation, which can highlight three essential parts: an adjustable system for the support of the arm and forearm (item 7), coupling systems for the two servo motors (items 5 and 1), and a grip handle for the patient’s pronation and supination (item 6).

A fixed base secured to the patient’s arm (item 1) and coupled to the other base of the forearm (item 3) and employing the rotation of the servomotor supports the orthosis prototype to perform the flexion and extension movements. The device has a section (item 4) that allows for the handle’s grip, making it possible to vary the arm’s position, allowing movements for different forearm lengths, and adding a length range of 7 cm. It is essential to mention that all foam covers the aluminum parts that have contact with the arm with neoprene in order to offer comfort and present a level of hygiene according to a product used in patients.

Now, following the guidelines proposed by the mechatronic design methodology, the simulation of static loads was performed to verify that each proposed mechanism did not suffer tension failures when applying a force, as shown in Table 3. For this purpose, the SolidWorks Simulation software was used, which estimates the Von Mises stress, a scalar that can indicate a good design for ductile materials, such as those used in [28].

The results presented in the table above show that, according to the study, the mechanisms do not present deformations or breaks in the materials when forces apply to them, demonstrating good behavior. Regarding the Von Mises stress analysis, the values approximate the elastic limit when bending and extension movements are performed, yielding a value of 1.630 × 10${}^{8}$ N/m${}^{2}$. Likewise, when analyzing the displacement results, upon inducing a maximum load on the mechanisms, it was observed that the most significant displacement occurred in the flexion and extension adaptations with a limit value of 7130 mm. Although the displacement is considerable, it does not present damage at the structural level in the prototype and is protected by a safety factor. It is essential to highlight that, for the design of the prototype, a safety factor of 2 was defined, ensuring that the adaptations do not present any rupture and the prototype has the correct functionality when applying a maximum force.

### 2.4. Sizing of the System Actuators

The central controller of the orthosis is the Arduino platform, which allows for the addition of multiple add-ons that provide flexibility in terms of actuators and sensors. Two 35 kg, high-torque Spt5435lv reference servo motors were used as the primary actuators of the system, which were selected for their accuracy, low-speed operation, minimal size, and a reduced mass compared to other motors. It also considered the motor’s torque since, according to Equation (1), the approximate mass that it can support is 2.92 kg, based on the mass range. The battery used for the servos is a 2-cell 7.4 v Lipo characterized by its reduced mass and the high amperage that it can provide to the servomotors.

It performed the control of the actuator movements of the system using pulse width modulation, where the time on high is equivalent to the servo angle or position. These values can vary and range from 0.5 to 1 millisecond for the 0${}^{\circ }$ position and from 2 to 2.4 milliseconds for the 180${}^{\circ }$ position; the signal period must be close to 20 milliseconds. At the level of the developed software, it implemented a delay function that allows for executing the movements at the physical level of the prototype and the healthcare professional. The healthcare professional sends the desired angles and the number of repetitions that he/she considers appropriate for the moment the patient is going through.

It is important to point out that the elbow rehabilitation aid system has an emergency stop that allows the process to be stopped. This functionality allows the patient to have some security when using the orthosis.

### 2.5. Graphical Interface for the Proposed System

The graphic interface is a fundamental part of the system since it is here where the health professional can interact with the prototype to determine angles and forces and consult the patient’s history. For the user interface programming, it was necessary to use a database where the patient’s data are stored and edited; Eclipse, XAMPP, and phpMyAdmin were used to develop the software. To make the connection between Arduino and NetBeans, the PanamaHitek library was used, which contains the methods to start and stop the connection, the serial communication parameters, and the functions to send and receive data [29].

The graphical user interface includes the requirements to properly program the rehabilitation by performing the flexion–extension and pronation–supination movements of the elbow. This interface allows the healthcare professional to establish the degrees of mobility required in that rehabilitation phase. It also includes a check box to select the number of repetitions that the user considers appropriate for the patient. Likewise, the patient’s default routine and the number of repetitions to be performed can be selected based on what the healthcare professional deems appropriate. Finally, the interface has two buttons that allow for the information to be sent once the user details it, thus starting the prototype and the rehabilitation movements, as shown in Figure 6A.

The second window of the interface (Figure 6B) allows the specialist to program the number of repetitions and the type of movement that the patient can perform, as well as to use the preloaded movements with the angle and number of repetitions, which can be executed daily during the period desired by the healthcare professional with the flexion–extension and prone–supination movements.

Along these lines, a web application was developed in PHP and linked to MySQL so that it can store patient data and the diagnosis of each patient (Figure 7). This element manages the client database for healthcare professionals, allowing them to create and delete records and export the data in various formats. On the web page, it is possible to edit and save data such as ID number, name, city, age, and diagnosis for each patient moving to the rehabilitation phase, as well as to manage the patients that the health professional has and their respective personalized rehabilitation phase.

### 2.6. Calibration of the System Actuators

In mechatronic design, the elements must have a correct synergy between design, simulation, and physical prototype, so concurrent engineering plays an important role in correcting errors at the time of construction. In this sense, to verify that the angle desired by the health professional is related to the one being executed, a comparison and characterization were made between the angle sent through the interface (measured value) and the accuracy of this movement with the help of a goniometer (actual value), thus finding the absolute error to determine the inaccuracy of the device using the following equation.

The physical therapists compared the servomotor’s angular position with a precision goniometer, calibrated in degrees to corroborate the servomotor’s angular position. As can be seen in Table 4, in the five tests performed with the device, the absolute error was minimal since, in the flexion–extension movements, it was 1%, which guarantees that the angular position of the servomotor complies with the same angle and that the absolute error in the pronation and supination mechanisms is 0%.

## 3. Results and Discussion

The main result of this research is a functional prototype of an orthosis to support one stage of the rehabilitation of the elbow joint (Figure 8). This device helps health professionals to treat pathologies through the execution of flexion–extension and prone–supination movements. The results show that it is a viable prototype validated through a case study, allowing us to establish that the mechatronic design helped to integrate the different areas of knowledge required for the construction and achievement of the device. In addition, the co-design methodology served as an integrating element to intertwine the proposed prototype with the software developed to support the system. Finally, some limitations were identified that could be corrected with new proposal iterations.

### Execution and Analysis of the Case Study

To assess the validity of the prototype, an exploratory case study was conducted (Figure 9) in an academic context, in which, ten people (with different masses and heights) were tested with the proposed device, and two healthcare professionals evaluated its performance and usability. Subsequently, the test subjects received a satisfaction survey to evaluate aspects such as the safety, hygiene, comfort, and functionality of the proposal, based on what it reported in [14]. Another survey was administered to the professionals to evaluate the proposal’s usability, acceptability, synergy, and performance based on their area of expertise. The experimental design is totally randomized. It has not diagnosed them with pathologies in the upper torso (healthy); during the test, they used the orthosis and performed different movements resembling a natural therapy when they present injuries in this body area.

The activities (Table 5) followed during the execution of the case study were: socialization and contextualization of the research project, presentation and introduction of the orthosis prototype, implementation of the orthosis in the study subjects, and, finally, the evaluation of the implementation through the use of a satisfaction survey to measure the usability, acceptability, synergy, and performance of the proposal.

The tabulation process of the survey was performed using the Likert scale, and it placed the size of the range of each quintile depending on each respondent. Thus, quintile one was placed as the respondents who showed the lowest ratings for the prototype, and, in quintile 5, were those who gave the highest rating or had the highest satisfaction in usability. In general, it can be mentioned that all respondents are in quintile five (Q5), with a score of 42 points out of 45, which represents 93.3% regarding the general acceptance of the proposed device.

Now, specifically, the mean scores of the respondents are represented by a line graph (Figure 10), which shows the existence of a positive trend in the ratings that are related to the evaluation of the overall performance of the device; in this sense, the evaluations show that 90% of the users stated that the orthosis is “very efficient” and the remaining 10% stated that it is “efficient”. This allows us to infer that, at the user level, the device had great acceptability and fulfilled the function for which it was built. Regarding the usability of the orthosis, the results show that the users felt comfortable with the proposal, where 65% of the respondents stated that the device was “very comfortable” while the remaining 35% rated it as “comfortable”. This shows that the device is pleasant and has materials that do not present contraindications for the surveyed users. Finally, the survey reflects that the respondents in the different items evaluated (comfort, hygiene, and functionality, among others) well accepted the prototype.

A survey was conducted to validate the prototype with two healthcare professionals along these lines. At a general level, the results show satisfaction, in which, 80% of the answers among all of the questions had the highest possible rating. Elements such as functionality, usability, the synergy between software and hardware, and device performance were evaluated with the highest score. Some essential elements in which the proposal had an average acceptance among the professionals were the learning curve of the system and the comfort of the system because the clinicians agreed that the orthosis and its operation (overall between hardware and software) need some adaptation time to understand the system and its constant interaction with the underlying technology. Likewise, at some points, the orthosis does not allow for comfortable postures when performing rehabilitation exercises, so these are elements that can be worked on in new iterations of the prototype, considering some guidelines of methodologies, such as industrial design [30].

Finally, among the comments made by users and healthcare professionals, it is mentioned that, although there is high acceptability towards the device, they recommend a different design for future iterations, where the prototype does not have such a robotic appearance but a more natural appearance, allowing it to adapt the appearance of an arm of the human body. Healthcare professionals were inclined to suggest new functionalities to the system, where elements such as rehabilitation routines preloaded by other professionals or the possibility of customizing them are functionalities that would be highly appreciated in the future, using software development techniques such as the so-called functionality repositories. They also show that new functionalities related to diagnostic support and not only to the treatment of pathologies could be added. For this, the angular sensors of the system and machine learning techniques could be used; for example, tests such as the internal rotation of elbow flexion [31] are diagnostic tests that could be evaluated directly by the system controller, following established parameters and introducing them into the artificial intelligence model, where parameters such as the force and acceleration with which the movement is performed could be evaluated, thus aiding in the diagnosis of diseases such as cubital tunnel syndrome or other conditions in the upper trunk.

## 4. Conclusions

This research presents the mechatronic design of a support system for the rehabilitation of the elbow by performing flexion–extension and pronation–supination movements with a graphical interface, which allows for the assisting of the rehabilitation of patients with conditions in the elbow area. The results suggest that the prototype assists the rehabilitation process by facilitating movements to help treat pathologies that reduce elbow mobility, achieving a very high level of satisfaction in both patients and health professionals. In addition, some limitations are identified in the device in terms of industrial design and some functionalities that can be addressed in future iterations.

The conjunction of methodologies such as mechatronic design, co-design, and IDEF0 are elements that allow for the obtaining of products with good synergy, and, as reflected in this study, when focused on perspectives of rehabilitation help in humans, can be a successful combination to obtain functional products, resulting in these methodologies being viable options for this application domain.

Finally, major future work includes improving the design of the prototype to make users feel more comfortable using it, while, at the level of functionalities, a diagnostic component could be added using technologies such as artificial intelligence or expert systems.

## Figures and Tables

**Figure 1 bioengineering-09-00287-f001:**
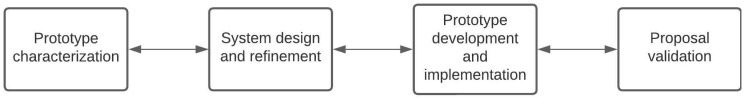
General activities followed for the design and implementation of the prototype rehabilitator for upper torso areas.

**Figure 2 bioengineering-09-00287-f002:**
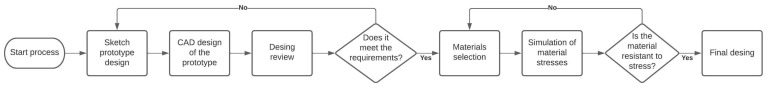
Methodology for the design and construction of the prototype.

**Figure 3 bioengineering-09-00287-f003:**
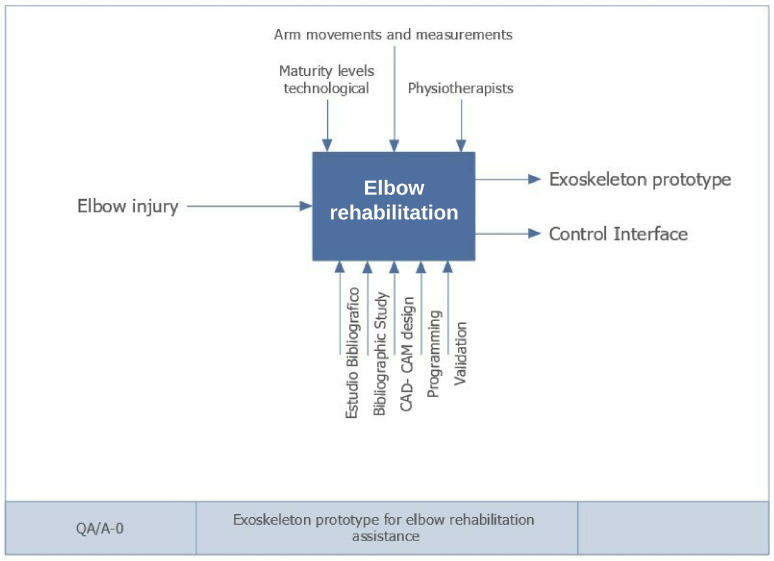
General system diagram IDEF A-0.

**Figure 4 bioengineering-09-00287-f004:**
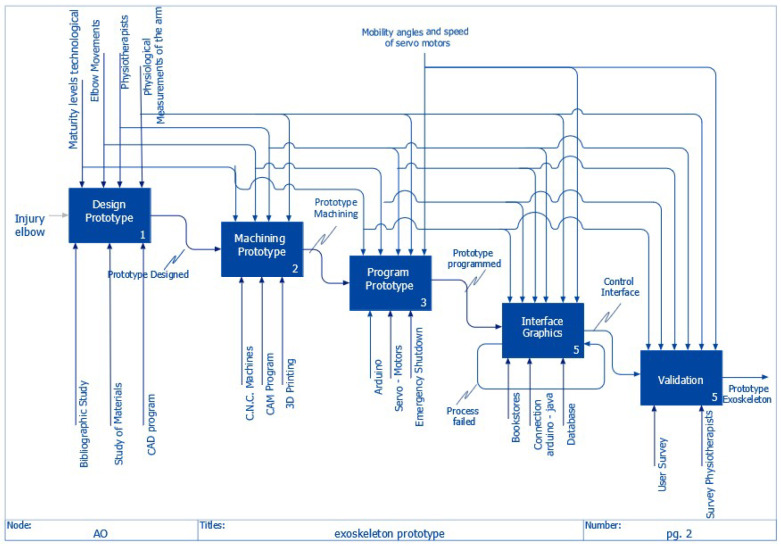
Detailed IDEFA0 system diagram (detail view).

**Figure 5 bioengineering-09-00287-f005:**
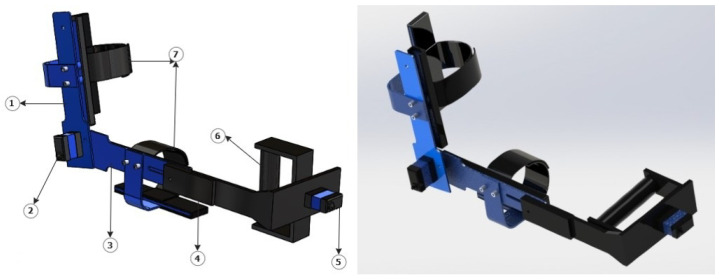
CAD design of an upper torso rehabilitative orthosis prototype.

**Figure 6 bioengineering-09-00287-f006:**
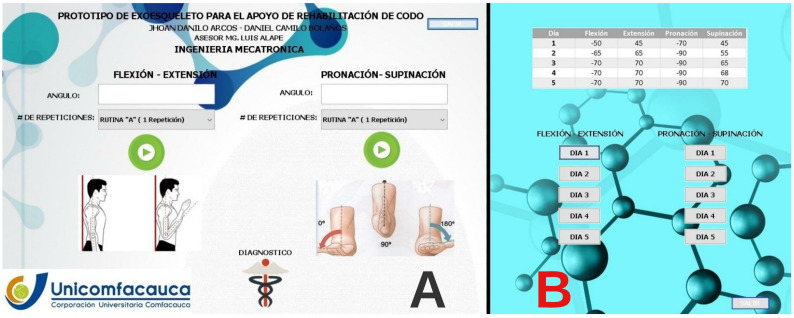
Graphical interface of the shoulder rehabilitation orthosis for the health care professional.

**Figure 7 bioengineering-09-00287-f007:**
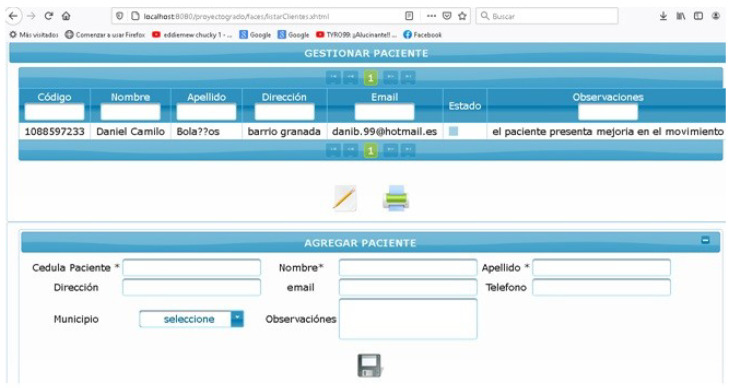
Graphical interface of the database of patients with upper torso pathologies.

**Figure 8 bioengineering-09-00287-f008:**
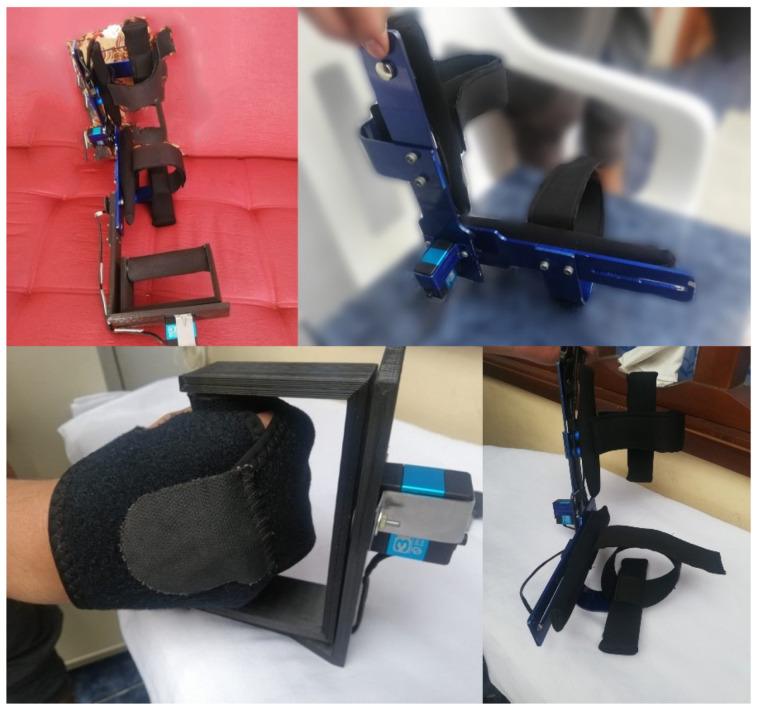
Photographic evidence of the orthosis prototype for upper torso rehabilitation.

**Figure 9 bioengineering-09-00287-f009:**
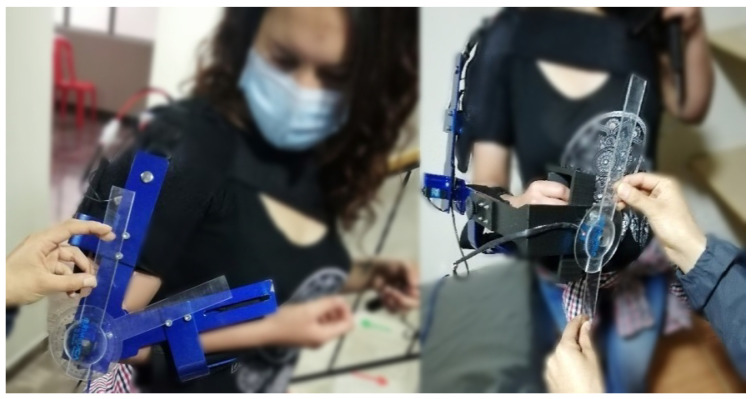
Prototype orthosis to support upper torso rehabilitation in case study.

**Figure 10 bioengineering-09-00287-f010:**
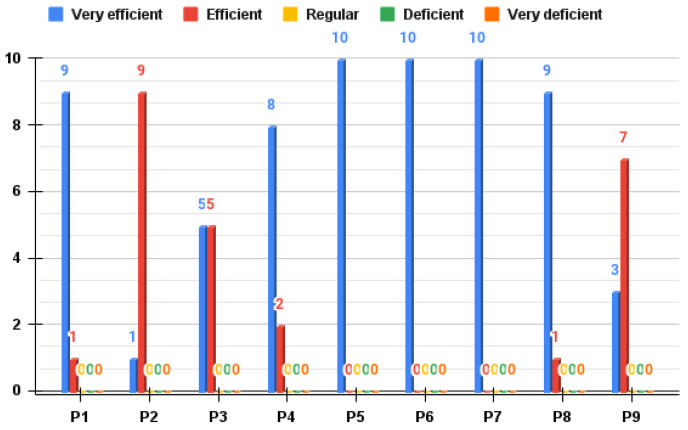
Results of the satisfaction survey for users of the upper torso rehabilitation support orthosis prototype.

**Table 1 bioengineering-09-00287-t001:** Normal elbow mobility values provided by the Association for the Study of Osteosynthesis and the American Academy of Orthopaedic Surgeons.

Motion	AO	AAOS
Flexion	0–150${}^{\circ }$	0–150${}^{\circ }$
Extension	0–10${}^{\circ }$	0${}^{\circ }$
Supination	0–90${}^{\circ }$	0–80${}^{\circ }$
Pronation	0–90${}^{\circ }$	0–80${}^{\circ }$

**Table 2 bioengineering-09-00287-t002:** Hand and forearm measurements and weights considered.

Range	Minimun [cm]	Maximun [cm]
Hand length	15.24	21.34
Hand width	6.86	9.91
Hand thickness	2.36	3.81
Handle diameter	3.18	3.81
Handgrip width	7.62	10.41
Fist length	9.65	14.22
Forearm length	21.08	27.43
Forearm diameter	6.1	8.13

**Table 3 bioengineering-09-00287-t003:** Stress analysis of the mechanisms of the prototype orthosis to support upper torso pathologies.

Mechanism	Tension of Von Mises [N/m2]	Displacement [mm]
Prone-supination of the wrist	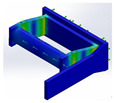 Elastic limit: 7.300 × 107 Value obtained: 7.697 × 105	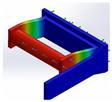 Minimum: 1 × 10−30 Maximum: 3.843 × 10−2
Prone-supination of the wrist	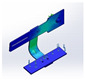 Elastic limit: 2.400 × 108 Value obtained: 1.630 × 105	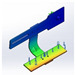 Minimum: 1 × 10−30 Maximum: 7.130

**Table 4 bioengineering-09-00287-t004:** Absolute error for the angles of movement measured with the goniometer and the proposed prototype.

Movement	Measured Value	Actual Value
Flexo-extension	70${}^{\circ }$	69${}^{\circ }$
Flexo-extension	135${}^{\circ }$	134${}^{\circ }$
Flexo-extension	20${}^{\circ }$	19${}^{\circ }$
pronation-supination	90${}^{\circ }$	90${}^{\circ }$
pronation-supination	120${}^{\circ }$	120${}^{\circ }$

**Table 5 bioengineering-09-00287-t005:** List of activities conducted in the case study.

Activities	Planned Duration	Support Tools
Socialization and contextualization of the research project	30 min	Presentation of the introduction to the case study and the conceptual elements of the proposal.
Presentation and introduction of the orthosis prototype	30 min	Document with the description of the prototype and presentation of the tool.
Implementation of the orthosis on the study subjects	45 min	Requirements document, orthosis guidance document, scenarios, and results templates for execution.
Evaluation of the implementation	20 min	Perception survey

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
