# Peer review of "Mechatronic Design of a Prototype Orthosis to Support Elbow Joint Rehabilitation"

_bioengineering, 2022, doi:10.3390/bioengineering9070287_

Round 1

Reviewer 1 Report

I prefer place quickly the major concerns about this manuscript before judging the details.

First, there are many more elbow exoskeletons, it seems to me that there is no novelty in the design and construction of this device. Please, underscore the novelty if there is any.

Second, it seems that there might be some interest in the experimental assessment of the exoskeleton, please, explain the experimental methods in the corresponding section. Also, it is suggested to separate the Results and the Discussion.

Third, what is the maximal torque that can apply the exoskeleton? Can it drive a human arm?

Fourth, if there are experiments with subjects, please indicate the reference from the corresponding Institutional Review Board and the completion of an informed consent form by all the participants.

Also, there is a figure (fig.9) that is in Spanish, this could be translated to English. I do not deem necessary to translate the pictures of the graphical interface, however, the meaning could be explained in the figure caption.

Author Response

Dear reviewers

Subject: Responses to the corrections made

Cordial greetings, I, at this moment, attach the corrections made to the article entitled " Mechatronic Design of a Prototype Orthosis to Support Elbow Joint Rehabilitation." In general, it can be mentioned that the entire article was improved grammatically, the bibliographical references were updated, and corrections were made regarding the author's guidelines. The corrections made according to each reviewer's comments are detailed below.

1. Comment: First, there are many more elbow exoskeletons; it seems that there is no novelty in the design and construction of this device. Please, underscore the novelty if there is any.

Answer: Accepted: A paragraph was added in the introduction, where the article's main contribution is explained. Therefore, it is considered that the main contribution of this research is to expand the area of knowledge through empirical evidence of a low-cost elbow rehabilitation prototype, built under the guidelines of mechatronic design and co-design methodology, resulting in an orthosis that can assist the stages of recovery of the mobility of the transverse elbow joint to any elbow pathology, through flexion and supination movements.

2. Second, it seems that there might be some interest in the experimental assessment of the exoskeleton; please, explain the experimental methods in the corresponding section. Also, it is suggested to separate the Results and the Discussion.

Answer: Partially accepted: We do not believe there is any interest in validating the proposal. The persons consulted are health professionals with experience in upper torso rehabilitation. A better description of the case study developed, together with the support instruments used, is presented in the results section (The instruments are accessible at the link at the end of the section).

3. Fourth, if there are experiments with subjects, please indicate the reference from the corresponding Institutional Review Board and complete an informed consent form by all the participants.

Answer: The experimental setting is entirely academic, although users signed a consent form. The university did not require an institutional review board to authorize the study. We also believe that the proposal is not sufficiently invasive for the board to be held and the time it would take to conduct a review board..

4. Also, there is a figure (fig.9) that is in Spanish, this could be translated to English. I do not deem necessary to translate the pictures of the graphical interface, however, the meaning could be explained in the figure caption.

Answer: Accepted: The figure in question has been modified and a better description of the graphic interface has been made.

Reviewer 2 Report

The following is the summary of the present manuscript:

    This research proposes the mechatronic design of an orthosis with a graphic interface that supports professionals in the rehabilitation of the elbow joint through the execution of flexion-extension and pronation-supination movements.Through the execution of a case study in a real environment, the device was validated, where the results suggest a functional and workable prototype that supports the treatment of pathologies in the elbow area, through the execution of the mentioned movements, supposing this a low-cost alternative, with elements to improve such as industrial design and new functionalities. The developed proposal shows potential as an economical product that health professionals can use.

   This is an interesting study, proposing a potential useful model of health practitioner. I have some comments for this manuscript:

First, the whole manuscript should be edited by a native English speaker. For example, there is no need to capitalize L, M and E for “lateral and medial epicondylitis.

Second, in Table 1, please provide the full term for AO and AAOS.

Third, please add more explanation for Figure 6. It seems that there are two subgraphs inside.

Fourth, Figure 7 should be enlarged. Otherwise, the words inside can not be visualized clearly.

Fifth, I suggest to separate “result” and “discussion” to fit for the regular format of a scientific paper.

Author Response

Popayán, Cauca.

18-May-2022

Dear reviewers

Subject: Responses to the corrections made

Cordial greetings, I, at this moment, attach the corrections made to the article entitled " Mechatronic Design of a Prototype Orthosis to Support Elbow Joint Rehabilitation." In general, it can be mentioned that the entire article was improved grammatically, the bibliographical references were updated, and corrections were made regarding the author's guidelines. The corrections made according to each reviewer's comments are detailed below.

1. Comment: First, the whole manuscript should be edited by a native English speaker. For example, there is no need to capitalize L, M and E for “lateral and medial epicondylitis.

Answer: The manuscript was generally revised by a native speaker and much of the sentence structure was modified.

2. Comment: Second, in Table 1, please provide the full term for AO and AAOS.

Answer: Accepted.

3. Comment: Third, please add more explanation for Figure 6. It seems that there are two subgraphs inside.

Answer: Accepted: Rewrote the graphical interface section for better explanation.

4. Comment: Fourth, Figure 7 should be enlarged. Otherwise, the words inside can not be visualized clearly.

Answer: Accepted: Image resized for a better understanding.

5. Comment:  Fifth, I suggest to separate “result” and “discussion” to fit for the regular format of a scientific paper.

Answer: Partially accepted: We do not believe that there is any interest in validating the proposal. The persons consulted are health professionals with experience in upper torso rehabilitation. A better description of the case study developed, together with the support instruments used, is presented in the results section (The instruments are accessible at the link at the end of the section).

Reviewer 3 Report

This research presents the mechatronic design of a new device for the rehabilitation of the elbow by performing movements with a graphical interface, which allows assisting the rehabilitation of patients with conditions
in the elbow area.

I think all  sections were written perfect.

Author Response

Dear reviewers

Subject: Responses to the corrections made

Cordial greetings, I, at this moment, attach the corrections made to the article entitled " Mechatronic Design of a Prototype Orthosis to Support Elbow Joint Rehabilitation." In general, it can be mentioned that the entire article was improved grammatically, the bibliographical references were updated, and corrections were made regarding the author's guidelines. The corrections made according to each reviewer's comments are detailed below.

1. Comment: I think all  sections were written perfect.
